

# Identification of a novel four-lncRNA signature as a prognostic indicator in cirrhotic hepatocellular carcinoma

Linkun Ma and Cunliang Deng

Department of Infectious Diseases, The Affiliated Hospital of Southwestern Medical University, Luzhou, China

## ABSTRACT

**Background**. Many studies have shown that long noncoding RNAs (lncRNA) are closely associated with the occurrence and development of various tumors and have the potential to be prognostic markers. Moreover, cirrhosis is an important prognostic risk factors in patients with liver cancer. Some studies have reported that lncRNA-related prognostic models have been used to predict overall survival (OS) and recurrence-free survival (RFS) in patients with hepatocellular carcinoma (HCC). However, no one has constructed a prognostic lncRNA model only in patients with cirrhotic HCC. Thus, it is necessary to screen novel potential lncRNA markers for improve the prognosis of cirrhotic HCC patients.

**Methods**. The probe expression profile dataset (GSE14520–GPL3921) from the Gene Expression Omnibus (GEO), which included 204 cirrhotic HCC samples, was reannotated and the lncRNA and mRNA expression dataset was obtained. The patients were randomly assigned to either the training set ($n = 103$) and testing set ($n = 100$). Univariate cox regression and the least absolute shrinkage and selection operator (LASSO) model were applied to screen lncRNAs linked to the OS of cirrhotic HCC in the training set. The lncRNAs having significant correlation with OS were then selected and the multivariate Cox regression model was implemented to construct the prognostic score model. Whether or not this model was related to RFS in the training set was simultaneously determined. The testing set was used to validate the lncRNA risk score model. A risk score based on the lncRNA signature was used for stratified analysis of different clinical features to test their prognostic performance. The prognostic lncRNA-related protein genes were identified by the co-expression matrix of lncRNA-mRNA, and the function of these lncRNAs was predicted through the enrichment of these co-expression genes.

**Results**. The signature consisted of four lncRNAs:AC093797.1,POLR2J4,AL121748.1 and AL162231.4. The risk model was closely correlated with the OS of cirrhotic HCC in the training cohort, with a hazard ratio (HR) of 3.650 (95% CI [1.761–7.566]) and log-rank $P$ value of 0.0002. Moreover, this model also showed favorable prognostic significance for RFS in the training set (HR: 2.392, 95% CI [1.374–4.164], log-rank $P = 0.0015$). The predictive performance of the four-lncRNA model for OS and RFS was verified in the testing set. Furthermore, the results of stratified analysis revealed that the four-lncRNA model was an independent factor in the prediction of OS and RFS of patients with clinical characteristics such as TNM (Tumor, Node, Metastasis system) stages I–II, Barcelona Clinic Liver Cancer (BCLC) stages 0–A, and solitary tumors in both the training set and testing set. The results of functional prediction showed that

Corresponding author
Cunliang Deng,
dengcunl@swmu.edu.cn,
dengcunl64@vip.sina.com

four lncRNAs may be potentially involve in multiple metabolic processes, such as amino acid, lipid, and glucose metabolism in cirrhotic HCC.

## INTRODUCTION

Worldwide, liver cancer remains among the top six prevalent carcinomas among males and females from Global Cancer Statistics with an estimated 841,080 new cases and 781,631 deaths occurring in 2018 (*International Agency for Research on Cancer, 2018*, http://gco.iarc.fr/today/fact-sheets-cancers). According to the US centers for disease control and prevention, the mortality of liver cancer rose by 25 percent (from 5.3 per 100,000 to 6.6 per 100,000) between 2006 and 2015 (*Centers for Disease Control and Prevention, 2018*). Despite the existing treatment methods, such as hepatectomy, liver transplantation, radiofrequency ablation, embolization therapy, and molecule-targeted chemotherapy, high mortality and recurrence rates of liver cancer, fundamentally, have not been changed, and new interventions to improve the poor prognosis for patients with liver cancer is in continuing demand (*Siegel, Miller & Jemal, 2018*). The stratified management of patients with hepatocellular carcinoma (HCC) according to specific clinical characteristics (e.g., using the Barcelona Clinic Liver Cancer staging system) has improved the prognosis of patients (*Villanueva, Hernandez-Gea & Llovet, 2013*).

Among the various pathological types of liver cancer, HCC is the most common, accounting for about 80% of all cases (*DeSantis et al., 2014*). The occurrence of HCC is closely related to liver fibrosis. Studies have reported that 80–90% of patients with HCC present liver fibrosis or cirrhosis (*El-Serag, 2011*). A recent meta-analysis of studies in cirrhotic patients with five liver diseases (hepatitis B, hepatitis C, primary biliary cholangitis, autoimmune hepatitis, and non-alcoholic steatohepatitis) showed that the incidence of liver cancer is significantly increased when liver disease develops into cirrhosis, indicating that cirrhosis is one of the crucial risk factors for HCC (*Tarao et al., 2019*). Patients with liver cancer, with or without cirrhosis, show different clinical characteristics, including differences in tumor size and prognostic factors (*Techathuvanan et al., 2015*). With the development of bioinformatics, many genomic biomarkers, including lncRNAs, have been explored. Three circulating lncRNAs, LINC00152, RP11-160H22.5 and XLOC014172, were found to could be candidate biomarkers towards diagnosis for hepatocellular carcinoma (*Yuan et al., 2017*). *Hu et al. (2017a)* meta-analysis showed that lncRNAs are correlated with the biological characteristics of gastric cancer, which may be a potential screening tool for the diagnosis of gastric cancer. *Hu et al. (2017b)* confirmed that lncRNA-SVUGP2 expression in hepatocellular carcinoma tissues was associated with patient prognosis. In addition, some lncRNAs have been found in serum to be biomarkers for tumor diagnosis. For instance, GIHCG has been found to be biomarkers for serum diagnosis of cervical cancer (*Zhang et al., 2019*). The lncRNAs are a type of noncoding RNAs longer than 200

nucleotides, which play various important roles in malignant tumors (*Ma et al., 2017*). The lncRNAs reportedly play a role in various biological processes, including cell proliferation, apoptosis, metastasis, and microvascular changes, and exhibit stem cell-like properties (*Yuan et al., 2012*; *Wang et al., 2014*; *Cui et al., 2015*; *Ma et al., 2016*). Several studies have demonstrated that lncRNAs contribute to the stratification of prognosis for patients with liver cancer (*Liao et al., 2018*; *Zhao et al., 2018*; *Gu et al., 2018*). For example, *Yan et al. (2019)* stratified the prognosis of HCC based on the seven-lncRNA signature, and the overall survival of the high-risk group was significantly lower than that of the low-risk group. Currently, a lncRNAs signature is absent for stratification of prognosis in patients with cirrhotic hepatocellular carcinoma. Some studies indicate that the knockdown of lncRNA SUMO1P3 can inhibit the growth and invasion of hepatocellular carcinoma and enhance its radiotherapy sensitivity (*Zhou et al., 2019*). This suggests that lncRNA can be a potential target for the treatment of hepatocellular carcinoma. Therefore, this study aims to further explore the prognostic biomarkers of the pathogenesis of cirrhotic HCC by analyzing microarray, and to provide some potential targets for the biological treatment of HCC.

## MATERIALS & METHODS

### Data availability statement

The following information was supplied regarding data availability: Raw data were generated in the study. All original data in this study were downloaded from the public databases GEO (https://www.ncbi.nlm.nih.gov/geo/). For the four-lncRNA signature identification, we used GSE14520.

### Probe reannotation and preparation of gene expression profiles

The GSE14520 expression profiles and clinical information analyzed in this study were derived from the GEO database (https://www.ncbi.nlm.nih.gov/geo/). To consider the expression and function of lncRNAs in the probes of HCC, the probes of the Affymetrix HT Human Genome U133A Array were reannotated to obtain the lncRNA and mRNA co-expression matrix using the following steps. Firstly, the Affymetrix HT Human Genome U133A Array probe was mapped to the gencode annotation file (Gencode.v29. transcripts, FASTA format,09/27/2018). Information about specific probe set annotation included the probe set ID, Ensemble ID, probe sequence, among other data. Secondly, the probe sets that were allocated to the Ensemble gene IDs in the gencode annotations were acquired. Human coding and noncoding gene annotation files (Homo_sapiens.GRCh38.95.chr.gtf.gz,GTF format,11/25/2018) from the Ensemble database were used to extract the matching information of gene ID and gene symbol. The Ensemble IDs of the probes were then assigned gene IDs, and only those annotated as "protein coding," "antisense," "sense intronic," "bidirectional promoter lncRNA," "lincRNA," "non-coding," "macro lncRNA," "processed transcript," "3′ overlapping ncRNA," and "sense overlapping" were retained. In addition, probes that corresponded to multiple Ensemble IDs were removed. Finally, 12,096 protein-coding transcripts and 610 lncRNA transcripts (different probe IDs may have corresponded to the same transcript) were reannotated.

The GSE14520 probe expression profile from the GEO database was annotated by the reannotated GPL3921 (Affymetrix HT Human Genome U133A Array), including 225 tumor samples and 220 paired non-tumor tissue samples. Among them, the survival information of 203 patients was obtained from the dataset of GSE14520. If multiple probes corresponded to the same gene, the final expression of the gene was determined by the arithmetic mean of multiple probes. The gene expression profile of GSE14520 was normalized by the limma software package. Normalization was performed using the normalizeBetweenArrays function in the limma package (*Bolstad et al., 2003*). The inclusion criteria were as follows: pathologically confirmed HCC tissues, and patients with liver cirrhosis, and complete follow-up survival information, including overall survival (OS) and recurrence-free survival (RFS). Exclusion criteria included non-tumor tissues, absence of histological examination results, pathological results indicating cholangiocarcinoma, hepatocholangiocarcinoma, or secondary liver cancer, and missing OS or RFS information.

It contained 225 HCC tissue samples and 220 adjacent non-tumor tissue samples. Out of 225 HCC samples, 221 with complete follow-up information (Survival status, Survival time, recurrence status and recurrence—free time) were included in the microarray. Of these 221 patients, 203 have cirrhosis. All 203 participants were patients who underwent radical resection of HCC at the Liver Cancer Institute and Zhongshan Hospital (Fudan University, Shanghai, China) between 2002 and 2003. Of the 203 participants, 177 were male and 26 were female. The mean age of all patients in the dataset was 51.3 years. Regarding the TNM stage of HCC, 153 of the 203 patients were in stage I or II, and 48 were in stage III. Only 2 patients had no documented cancer stage.

## Data and clinical characteristics

All 204 patients with liver cancer from the GSE14520 of the GEO database had complete survival information and were included in the study. These patients were randomly divided into the training set ($n = 103$) and testing set ($n = 100$) using the R caret software package. The clinical parameters of the two groups of patients are presented in Table 1.

## Construction of lncRNA expression-based prognostic signature

Univariate Cox regression analysis was used to screen for significant prognostic lncRNAs in the training set. A *P* value <0.05 was selected as the threshold for candidate lncRNAs. Candidate prognostic lncRNAs were further reduced by least absolute shrinkage and selection operator (LASSO) regression. The principle of LASSO regression is to eliminate some variables through penalty rules, and ultimately leave all potential predictors with a non-zero coefficient (*Gao, Kwan & Shi, 2010*). The penalty argument lambda was determined by the cross-validation method using the R glmnet software package. Lambda.min, the lambda value corresponding to the minimum value of the cross-validation error mean, was identified to determine the potential prognostic lncRNAs (*Tibshirani, 1997*). Each "significant" lncRNA obtained in the above steps were then fitted into a multivariate Cox regression model, and lncRNAs associated with independent prognostic criteria were selected. The prognostic risk formulas were formed, based on the expression level of those remaining eligible lncRNAs multiplied by the multivariate Cox regression

**Table 1 Clinical parameters of patients in the training set and testing set.**

| Variables | Training set (*n* = 103) | Testing set (*n* = 100) | *P* value |
|---|---|---|---|
| Gender | | | |
| Female | 13 | 13 | 1 |
| Male | 90 | 87 | |
| Age | | | |
| <60 | 80 | 81 | 0.68 |
| ≥60 | 23 | 19 | |
| ALT | | | |
| ≤50 U/L | 61 | 54 | 0.542 |
| >50 U/L | 42 | 46 | |
| Main tumor size | | | |
| ≤5 cm | 68 | 58 | |
| >5 cm | 34 | 42 | 0.26 |
| NA | 1 | 0 | |
| Multinodular | | | |
| NO | 86 | 74 | 0.138 |
| YES | 17 | 26 | |
| TNM stage | | | |
| I+II | 82 | 71 | |
| III | 21 | 27 | 0.305 |
| NA | 0 | 2 | |
| BCLC stage | | | |
| 0+A | 83 | 69 | |
| B+C | 20 | 29 | 0.13 |
| NA | 0 | 2 | |
| AFP | | | |
| ≤300 ng/ml | 54 | 52 | |
| >300 ng/ml | 46 | 48 | 0.887 |
| NA | 3 | 0 | |

**Notes.**
BCLC stage, Barcelona Clinic Liver Cancer stage; Multinodular, Whether the tumor has multiple nodules.

coefficient. Risk scores for each sample were calculated using the risk score formula, and patients were divided into high-risk and low-risk groups based on the median risk score of the training cohort as the cutoff value. The same formulas and cutoff values were used to calculate the risk score and risk grouping for patients in the testing set (*Zhao et al., 2018*). A Kaplan–Meier survival curve and log-rank test were used to analyze the overall survival (OS) and recurrence-free survival (RFS) of patients in the high-risk and low-risk groups in both the training and testing set (*Li et al., 2018*). A receiver operating characteristic (ROC) curve plotted with the survivalROC software package (version 1.0.3) was adopted to evaluate the specificity and sensitivity of the survival prediction (*Heagerty, Lumley & Pepe, 2000*). Both ROC and Kaplan–Meier survival curves were constructed with the R studio software (version 3.5.1). A *P* value < 0.05 was deemed statistically significant.

## Functional enrichment analyses

In order to obtain the lncRNA-mRNA co-expression pair, we conducted Pearson correlation analyses between the four lncRNAs identified and the expression profile of the protein-coding genes of the discovery set. The protein-coding genes with a correlation coefficient >0.4 and $P < 0.001$ were recognized as the four lncRNA-associated genes. The correlated mRNAs were evaluated by functional enrichment analysis to explore the functions of the four prognostic lncRNAs using the clusterProfiler (version 3.10.1) and org.Hs.eg.db (version 3.7.0) package (*Yu et al., 2012*). Significantly enriched Gene Ontology (GO) terms and Kyoko Encyclopedia of Genes and Genomes (KEGG) pathways with a $P$ value <0.05 and $q$ value <0.05 were visualized using the R studio (version 3.5.1) software.

## Statistical analysis

The prognostic ability of the four-lncRNA signature under different clinical parameters was analyzed by Kaplan–Meier survival analysis, to determine which population is most suited for this prognostic model. The survival curve was drawn using the R survminer software package. Statistical analyses were conducted using the SPSS software (version 24.0) and R studio (version 3.5.1).

# RESULTS

## Construction of risk prognostic scoring system in the training set

The exploration process of this study is shown in Fig. 1. Firstly, 610 lncRNAs were initially screened in the training set (Table S1), and 35 potential OS-related lncRNAs were obtained by a univariate Cox risk regression model (Table S2) ($P < 0.05$). The remaining lncRNAs were further selected using LASSO regression analysis, and cross-validation was used to select the penalty parameters (Figs. 2A–2B). Five lncRNAs were identified by lambda.min value (Table S3). The lncRNAs obtained in the above steps were inserted into the multivariate Cox regression model. The expression values of four independent prognostic lncRNAs ($P$ value <0.05) and their correlation coefficients in a multivariate regression model were then used to construct prognostic signatures. Detailed information and the significance of survival prediction by the four lncRNAs are presented in Table 2.

Risk score = (expression quantity of AC093797. 1× -0.4818) + (expression quantity of AL121748. 1× 0.4404) + (expression quantity of AL162231. 4× 1.2845) + (expression quantity of POLR2J 4× −1.5170).

The risk score formula in the training set was used to calculate the risk score of each patient. The median score for all patients in the training set was deemed the cutoff value, and facilitated the division of patients into a high-risk group ($n = 51$) and low-risk group ($n = 52$) (Table S4). Patients of the training group were ranked in ascending order of risk score (Fig. 3A). Moreover, the expression profiles of the four lncRNAs were plotted on a heatmap (Fig. 3B). In addition, a scatter plot was constructed to show the OS status and recurrence status of patients (Figs. 3C–3D). The high-risk group had a worse prognosis than the low-risk group in terms of OS and RFS. The prognosis of the high-risk group was compared with that of the low-risk group using Kaplan–Meier curve analysis. The

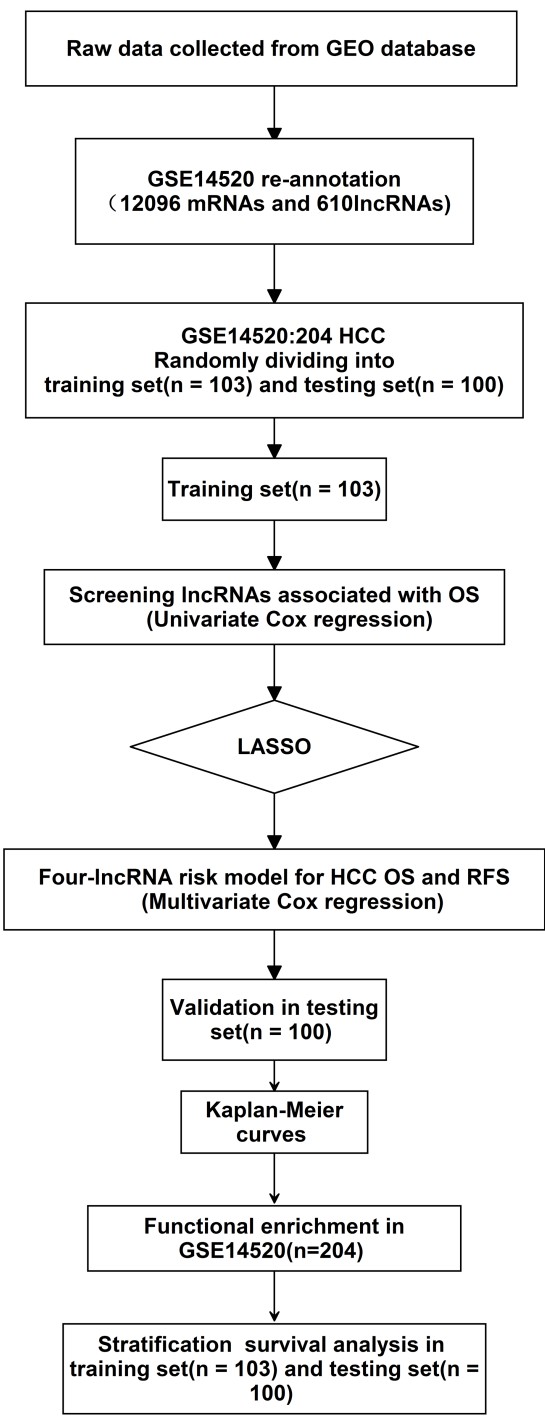

**Figure 1  Analysis of flowchart.** The flowchart indicates the exploration process and potential mechanism of cirrhotic HCC prognostic lncRNAs.

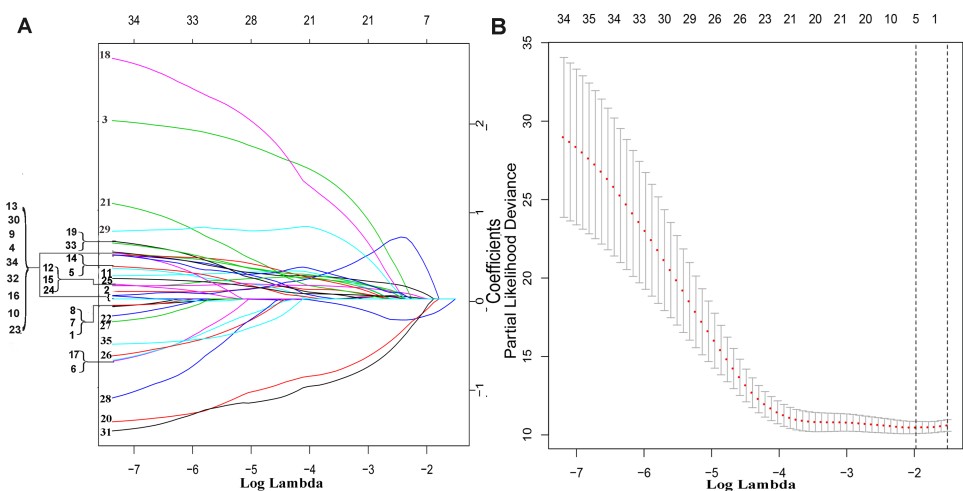

**Figure 2** **The remaining lncRNAs were further selected using LASSO regression analysis, and cross-validation was used to select the penalty parameters.** (A) LASSO coefficient profiles of the 35 candidate lncRNAs. Ten-time cross-validation for penalty parameters selection in the LASSO model. (B) A vertical line is drawn at the value chosen by 10-fold cross-validation.

**Table 2** **Five candidate lncRNAs screened by LASSO regression were inserted in multivariate COX regression.** Four lncRNAs with independent prognostic significance for overall survival were screened out. Details of the four lncRNAs are as follows.

| Gene name | Ensemble ID | coefficient | Hazard ratio | P value | Chromosome |
|-----------|-------------|-------------|--------------|---------|------------|
| AC093797.1 | ENSG00000233110 | −0.4818 | 0.6177 | 0.0003 | Chr4:185587909-185594004(+) |
| AL121748.1 | ENSG00000238258 | 0.4404 | 1.5534 | 0.038 | Chr10:33211277-33213805(+) |
| AL162231.4 | ENSG00000261215 | 1.2845 | 3.6129 | 0.0221 | Chr9:34661903-34666029(-) |
| POLR2J4 | ENSG00000214783 | −1.517 | 0.2194 | 0.0081 | Chr7:43940895-44019175(-) |

**Notes.**
Correlation coefficient and Hazard ratio were the results of multivariate COX regression.

results showed that the OS (HR: 3.650, 95% CI [1.761–7.566], log-rank $P = 0.0002$) and RFS (HR: 2.392, 95% CI [1.374–4.164], log-rank $P = 0.0015$) of high-risk patients were significantly lower than those of low-risk patients (Figs. 4A–4B). Furthermore, according to the four-lncRNA signature, the area under the curve of time-dependent ROC analysis of the 5-year OS and RFS predicted in the training set reached 0.839 and 0.715, respectively (Figs. 4C–4D).

## Prognostic performance of the four-lncRNA signature in the testing set

To further assess the prognostic significance of the four-lncRNA signature, internal validation was performed in a group of 100 patients from the testing set. The established formula for the training set was used to calculate the patient risk score for the testing set. The same cutoff value used for the training set was also used to divide the testing set into high-risk ($n = 44$) and low-risk cohorts ($n = 56$) (Table S5). Kaplan–Meier survival analysis was also performed in the testing set. As shown in Figs. 5A–5B, the four-lncRNA signature showed favorable prognostic value in differentiating the risk stratification of death (HR:

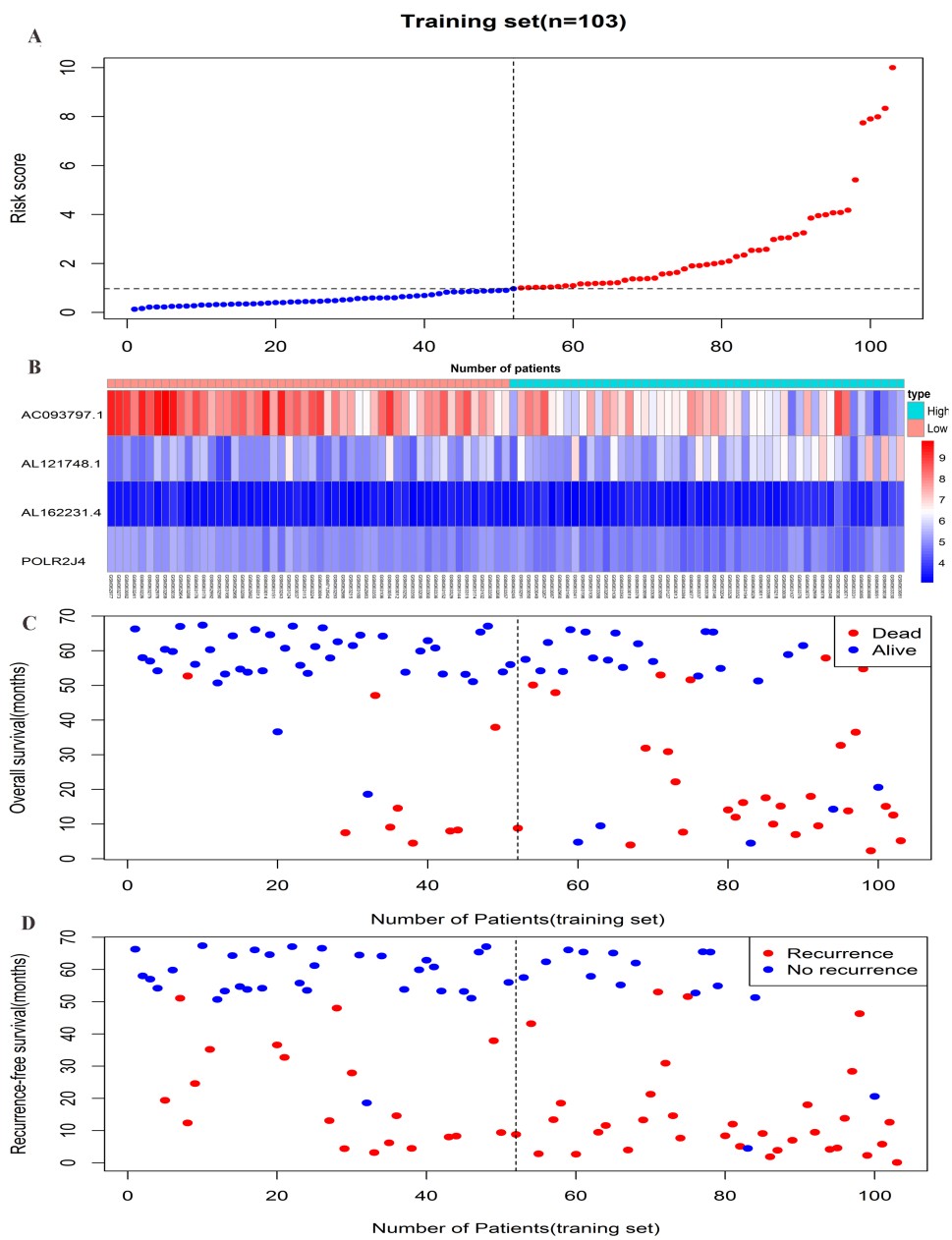

**Figure 3** **Prognostic efficiency of the prognostic risk scoring system for overall survival and recurrence-free survival.** (A) The four-lncRNA risk score distribution. (B) The heatmap of four lncRNAs gene expression in the high-risk and low-risk groups for the training set. (C) Overall survival status of patients in training set. (D) Recurrence status of patients in training set.

2.475, 95% CI [1.347–4.547], log-rank $P = 0.0015$) and recurrence (HR: 2.245, 95% CI [1.349–3.736], log-rank $P = 0.0014$) in patients. The ROC curve was used to analyze the predictive power of risk models in the validation group for 5-year OS (Fig. 5C) and RFS (Fig. 5D). The OS and RFS of patients in the testing group were plotted (Figs. 5E–5F).

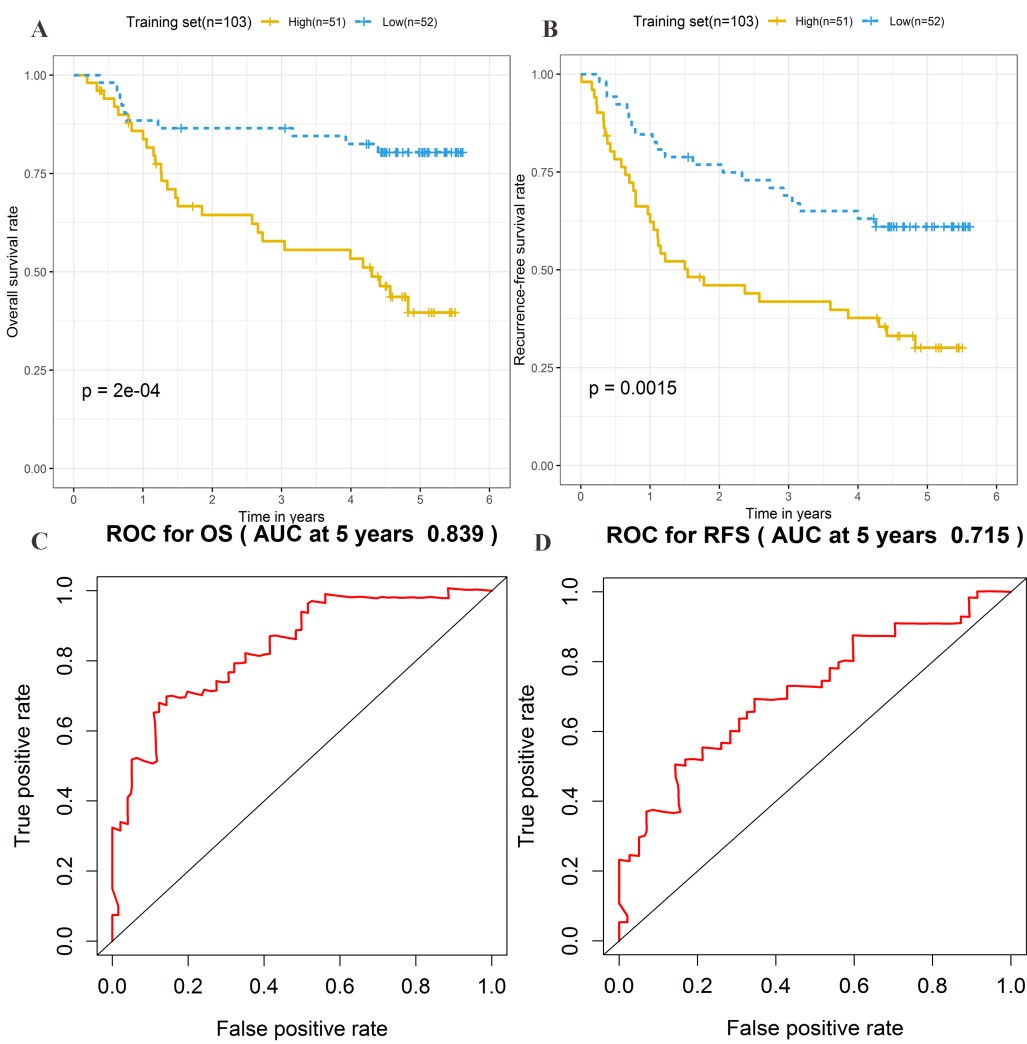

**Figure 4** **Prognostic significance evaluation of four-lncRNA signature for overall survival (OS) and recurrence-free survival (RFS) in the training set.** Kaplan–Meier analysis for high-risk and low-risk groups was used to analyze OS and RFS. (A) The survival curve of OS. (B) The survival curve of RFS. (C) ROC curve for prognosis prediction of the risk score model at 5 years of OS. (D) ROC curve at 5 years of RFS.

These results suggested that the four-lncRNA signature in patients with cirrhotic HCC has considerable potential in predicting OS and RFS.

## Stratification analysis

To explore the different clinical characteristics for which the four-lncRNA model is applicable, we conducted subgroup analyses of OS and RFS on patients of both the training set and testing set. The risk grouping principle was based on the median four-lncRNA risk score of the training set. Subgroup analyses combining the training and validation sets showed that the risk score model based on the four-lncRNA signature was more appropriate for patients with the following characteristics: BCLC stages 0–A, solitary

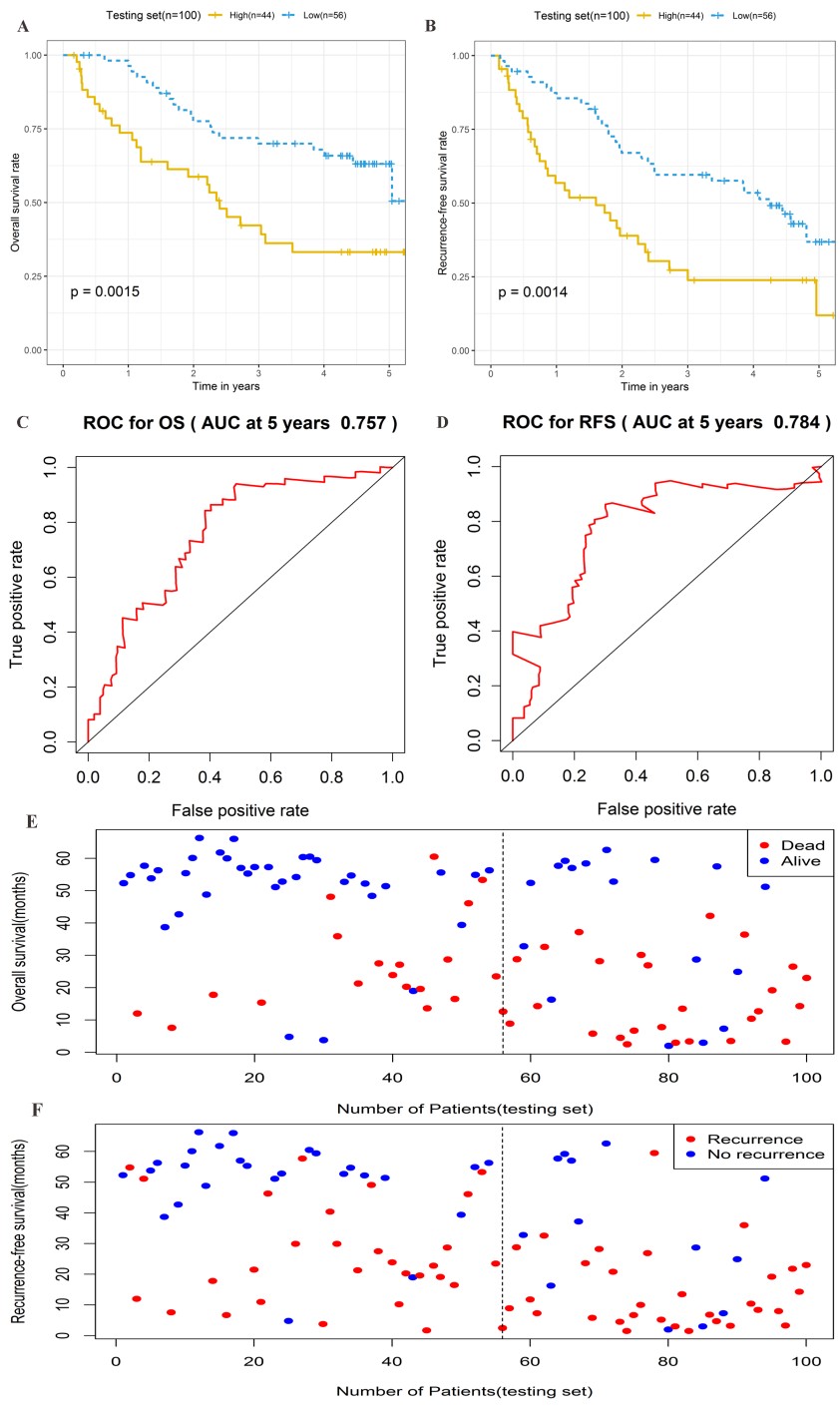

**Figure 5** **Prognostic performance of four-lncRNA signature on overall survival(OS) and recurrence-free survival(RFS) in the testing set.** Kaplan–Meier analysis for high-risk and low-risk groups was used to analyze overall survival and recurrence-free survival. (A) The survival curve of OS. (B) The survival curve of RFS. (C) ROC curve for prognosis prediction of the risk score model at 5 years of OS. (D) ROC curve at 5 years of RFS. (E) Overall survival status of patients in testing set. (F) Recurrence status of patients in testing set.

**Table 3** Stratified analysis of the overall survival of four-lncRNA signature in the training set and testing set.

| Variable | Training set | | | Testing set | | |
|---|---|---|---|---|---|---|
| | Number (High/Low) | HR (95% CI) | P value | Number (High/Low) | HR (95% CI) | P value |
| Gender | | | | | | |
| Female | 6/7 | 3.939 (0.409–37.96) | 0.2 | 6/7 | 4.037 (0.365–44.68) | 0.22 |
| Male | 45/45 | 3.603 (1.668–7.781) | 0.0005* | 38/49 | 2.405 (1.312–4.408) | 0.0034* |
| Age | | | | | | |
| <60 | 45/35 | 3.536 (1.512–8.27) | 0.0019* | 34/47 | 2.412 (1.269–4.584) | 0.0056* |
| ≥60 | 6/17 | 4.253 (0.946–19.11) | 0.059 | 10/9 | 7.153 (0.858–59.62) | 0.069 |
| ALT | | | | | | |
| ≤50 U/L | 27/34 | 2.985 (1.325–6.726) | 0.0056* | 22/32 | 2.479 (1.047–5.868) | 0.033* |
| >50 U/L | 24/18 | 10.87 (1.386–85.23) | 0.0045* | 22/24 | 2.358 (1.056–5.269) | 0.032* |
| Main Tumor Size | | | | | | |
| ≤5 cm | 32/36 | 5.949 (1.963–18.02) | 0.00036* | 21/37 | 2.145 (0.993–4.637) | 0.0523 |
| >5 cm | 18/16 | 2.213 (0.825–5.94) | 0.11 | 23/19 | 2.5 (0.992–6.298) | 0.0519 |
| Multinodular | | | | | | |
| NO | 39/47 | 4.157 (1.834–9.419) | 0.00021* | 30/44 | 3.017 (1.472–6.183) | 0.0015* |
| YES | 12/5 | 1.575 (0.316–7.857) | 0.58 | 14/12 | 0.641 (0.231–1.777) | 0.39 |
| TNM stage | | | | | | |
| I+II | 37/45 | 4.186 (1.633–10.73) | 0.0012* | 24/47 | 2.779 (1.337–5.778) | 0.0043* |
| III | 14/7 | 1.778 (0.555–5.693) | 0.33 | 19/8 | 1.454 (0.503–4.206) | 0.49 |
| BCLC stage | | | | | | |
| 0+A | 36/47 | 4.728 (1.858–12.03) | 0.00033* | 26/43 | 3.238 (1.481–7.08) | 0.0019* |
| B+C | 15/5 | 0.879 (0.275–2.816) | 0.83 | 17/12 | 0.630 (0.252–1.575) | 0.32 |
| AFP | | | | | | |
| ≤300 ng/ml | 21/33 | 3.944 (1.446–10.76) | 0.0039* | 17/35 | 4.625 (1.842–11.61) | 0.00037* |
| >300 ng/ml | 29/17 | 3.012 (1.004–9.033) | 0.039* | 27/21 | 1.438 (0.659–3.136) | 0.36 |

Notes.
*Statistically significant.
Abbreviations: BCLC stage, Barcelona Clinic Liver Cancer stage; Multinodular, whether the tumor has multiple nodules; HR, hazard ratio; 95% CI, 95% confidence interval.

tumors, and Tumor, Node, Metastasis system (TNM) stages I–II. The results of subgroup analyses of the OS and RFS are shown in Tables 3 and 4, respectively. Kaplan–Meier curves were constructed for both high-risk and low-risk patients with TNM stages I–II, solitary tumors, and BCLC stages 0–A, to analyze the prognostic value of risk models in OS (Figs. 6A–6F) and RFS of cirrhotic HCC (Figs. 7A–7F). Among the patients with the three aforementioned clinical characteristics, the OS (Fig. 6A ($P = 0.00021$), Fig. 6B ($P = 0.0015$), Fig. 6C ($P = 0.0012$), Fig. 6D ($P = 0.0043$), Fig. 6E ($P = 0.00033$), and Fig. 6F ($P = 0.0019$)) and RFS (Fig. 7A ($P = 0.0018$), Fig. 7B ($P = 0.0019$), Fig. 7C ($P = 0.0028$), Fig. 7D ($P = 0.00061$), Fig. 7E ($P = 0.0023$), and Fig. 7F ($P = 0.0018$)) of high-risk patients were worse than those of low-risk patients.

## Functional enrichment analyses

Firstly, the co-expression matrix of the four lncRNAs and 12,100 protein-coding genes were extracted from 204 patients with tumor tissue gene expression profile data from GSE14520.

**Table 4  Stratified analysis of the recurrence-free survival of four-lncRNA signature in the training set and testing set.**

| Variable | Training set | | | Testing set | | |
|---|---|---|---|---|---|---|
| | Number (High/Low) | HR (95% CI) | P value | Number (High/Low) | HR (95% CI) | P value |
| Gender | | | | | | |
| Female | 6/7 | 5.765 (0.64–51.95) | 0.078 | 6/7 | 7.811 (0.801–76.15) | 0.077 |
| Male | 45/45 | 2.166 (1.217–3.854) | 0.0071[*] | 38/49 | 2.072 (1.224–3.508) | 0.0056[*] |
| Age | | | | | | |
| <60 | 45/35 | 2.34 (1.214–4.51) | 0.0089[*] | 34/47 | 1.954 (1.102–3.464) | 0.02[*] |
| ≥60 | 6/17 | 3.758 (1.149–12.29) | 0.019[*] | 9/9 | 4.428 (1.19–16.47) | 0.015[*] |
| ALT | | | | | | |
| ≤50 U/L | 27/34 | 2.164 (1.08–4.333) | 0.026[*] | 22/32 | 1.974 (0.947–4.115) | 0.064 |
| >50U/L | 24/18 | 2.852 (1.102–7.379) | 0.024[*] | 22/24 | 2.394 (1.179–4.86) | 0.013[*] |
| Main Tumor Size | | | | | | |
| ≤5 cm | 32/36 | 3.655 (1.725–7.745) | 0.0003[*] | 21/37 | 1.95 (1.018–3.735) | 0.04[*] |
| >5 cm | 18/16 | 1.324 (0.571–3.07) | 0.51 | 23/19 | 2.586 (1.079–6.199) | 0.028[*] |
| Multinodular | | | | | | |
| NO | 39/47 | 2.514 (1.382–4.573) | 0.0018[*] | 30/44 | 2.516 (1.379–4.592) | 0.0019[*] |
| YES | 12/5 | 1.596 (0.329–7.749) | 0.56 | 14/12 | 1.658 (0.635–4.331) | 0.3 |
| TNM stage | | | | | | |
| I+II | 37/45 | 2.608 (1.357–5.01) | 0.0028[*] | 24/47 | 2.837 (1.524–5.282) | 0.00061[*] |
| III | 14/7 | 1.214 (0.42–3.512) | 0.72 | 19/8 | 1.167 (0.428–3.184) | 0.77 |
| BCLC stage | | | | | | |
| 0+A | 36/47 | 2.658 (1.385–5.104) | 0.0023[*] | 26/43 | 2.671 (1.41–5.058) | 0.0018[*] |
| B+C | 15/5 | 0.564 (0.187–1.7) | 0.3 | 17/12 | 1.608 (0.668–3.872) | 0.29 |
| AFP | | | | | | |
| ≤300 ng/ml | 21/33 | 2.228 (1.073–4.625) | 0.027[*] | 17/35 | 3.615 (1.692–7.724) | 0.00042[*] |
| >300 ng/ml | 29/17 | 2.489 (0.991–6.255) | 0.0524 | 27/21 | 1.426 (0.704–2.89) | 0.32 |

**Notes.**
[*]Statistically significant.

Abbreviations: BCLC stage, Barcelona Clinic Liver Cancer stage; Multinodular, whether the tumor has multiple nodules; HR, hazard ratio; 95%CI, 95% confidence interval.

The correlation between lncRNAs and protein-coding genes was analyzed. The significantly correlated protein-coding genes (Pearson coefficient >0.4, $P < 0.001$) were used for GO analysis (Table S6) and KEGG enrichment analysis (Table S7) to determine the potential mechanism of the four lncRNAs in regulating HCC. Functional enrichment analysis was conducted with the R org.Hs.eg.db (version 3.7.0) and clusterProfiler (version 3.10.1) package (*Yu et al., 2012*). The first 25 significant GO terms ($P < 0.05$, $q < 0.05$) (Fig. 8A) and first 25 significant KEGG pathways ($P < 0.05$, $q < 0.05$) (Fig. 8B) were charted using the R software (version 3.5.1). The results of GO analysis revealed that four-lncRNA-related functions showed enrichment in protein-coding genes involved in a large group of GO terms, including those associated with transmembrane transport, redox reactions, and fatty acid metabolism. Furthermore, KEGG analysis showed significant associations with a series of enriched pathways, including the peroxisome proliferator-activated receptor

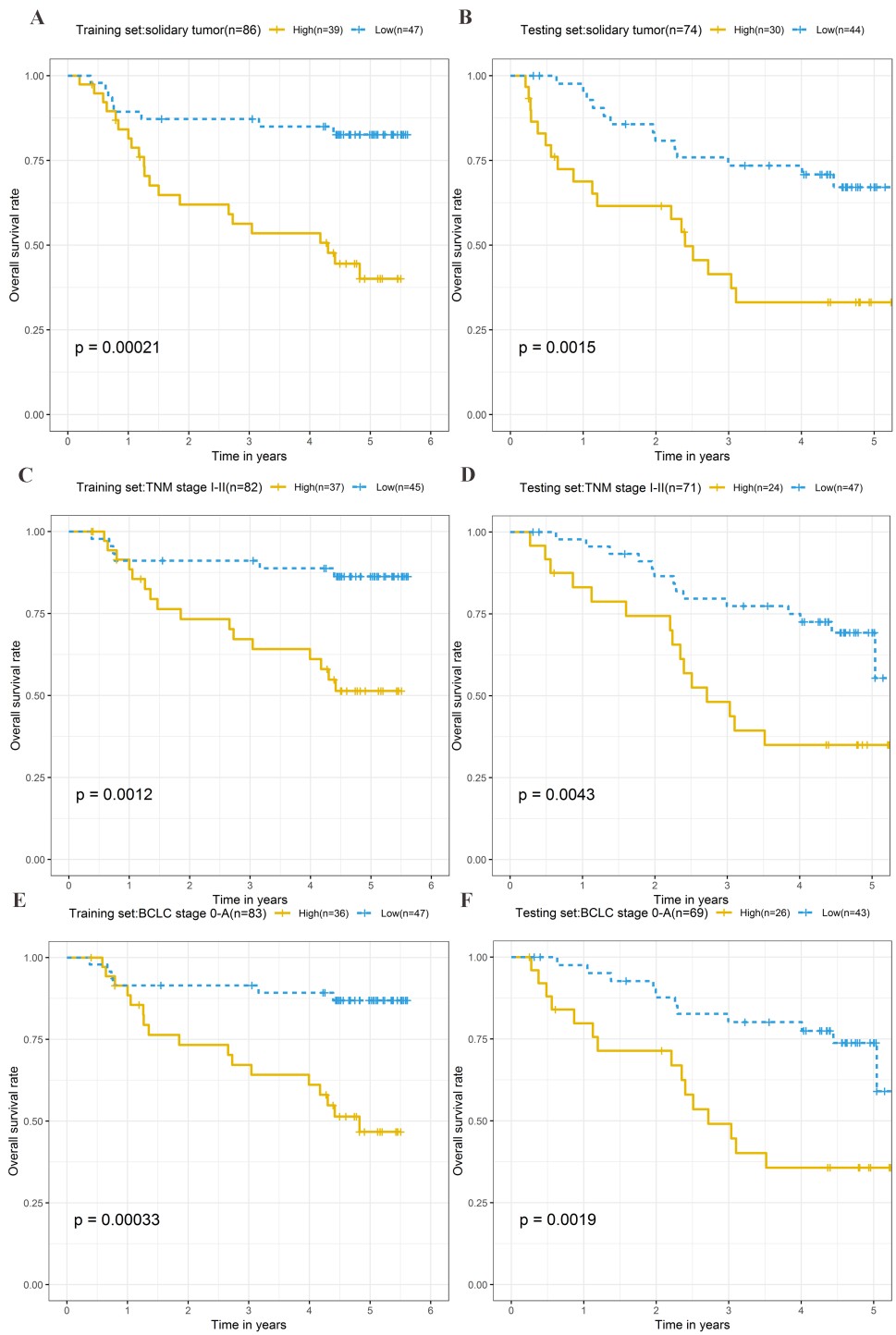

**Figure 6** **Kaplan–Meier curves of the overall survival for high-risk and low-risk patients with different clinical characteristics in training set and testing set.** Following clinical characteristics: (A) solitary tumor in training set. (B) solitary tumor in testing set. (C) TNM stage I-II in training set. (D) TNM stage I-II in testing set. (E) BCLC stage 0-A in training set. (F) BCLC stage 0-A in testing set.

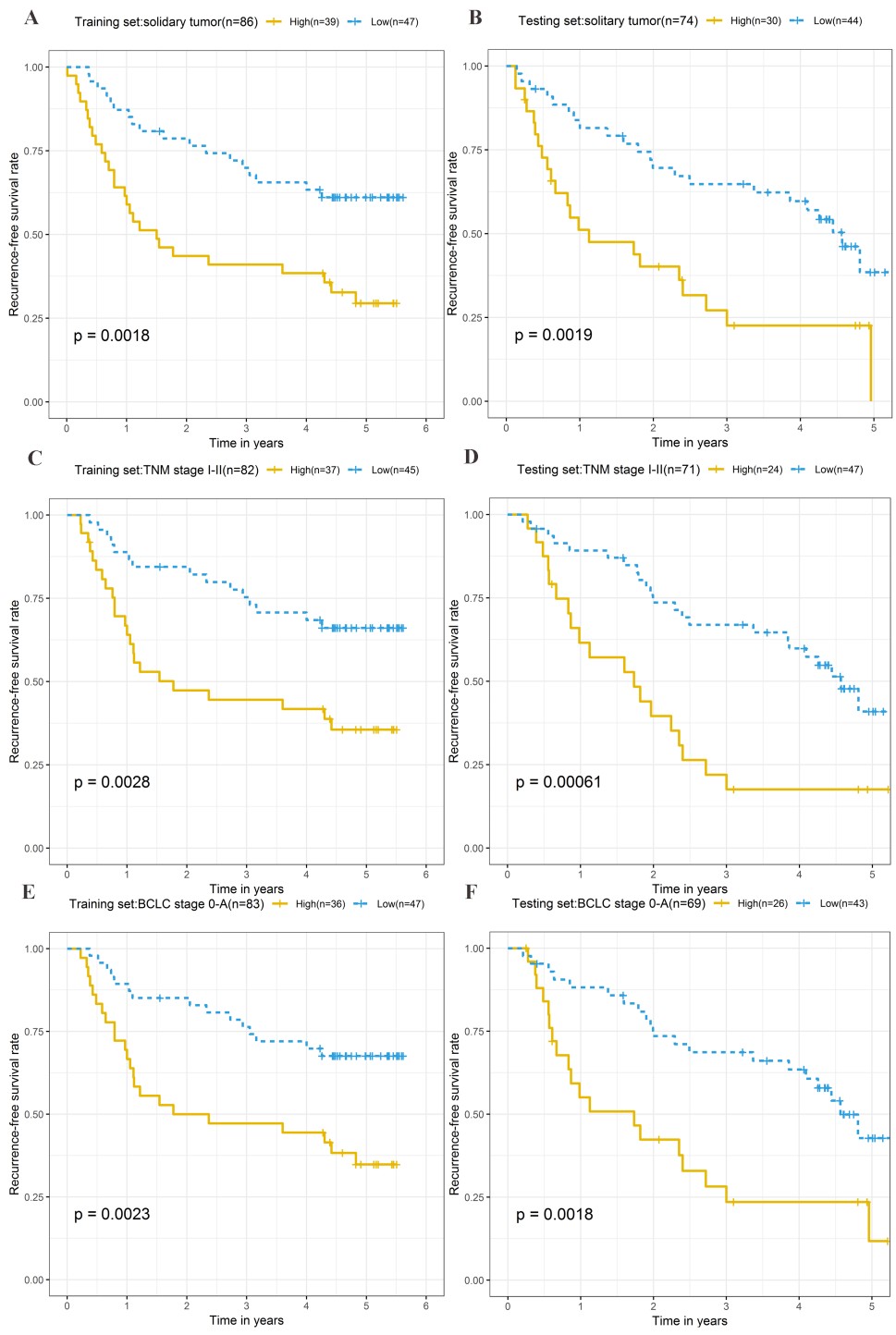

**Figure 7 Kaplan–Meier curves of the recurrence-free survival for high-risk and low-risk patients with different clinical characteristics in training set and testing set.** Following clinical characteristics: (A) solitary tumor in training set. (B) solitary tumor in testing set. (C) TNM stage I-II in training set. (D) TNM stage I-II in testing set. (E) BCLC stage 0-A in training set. (F) BCLC stage 0-A in testing set.

 

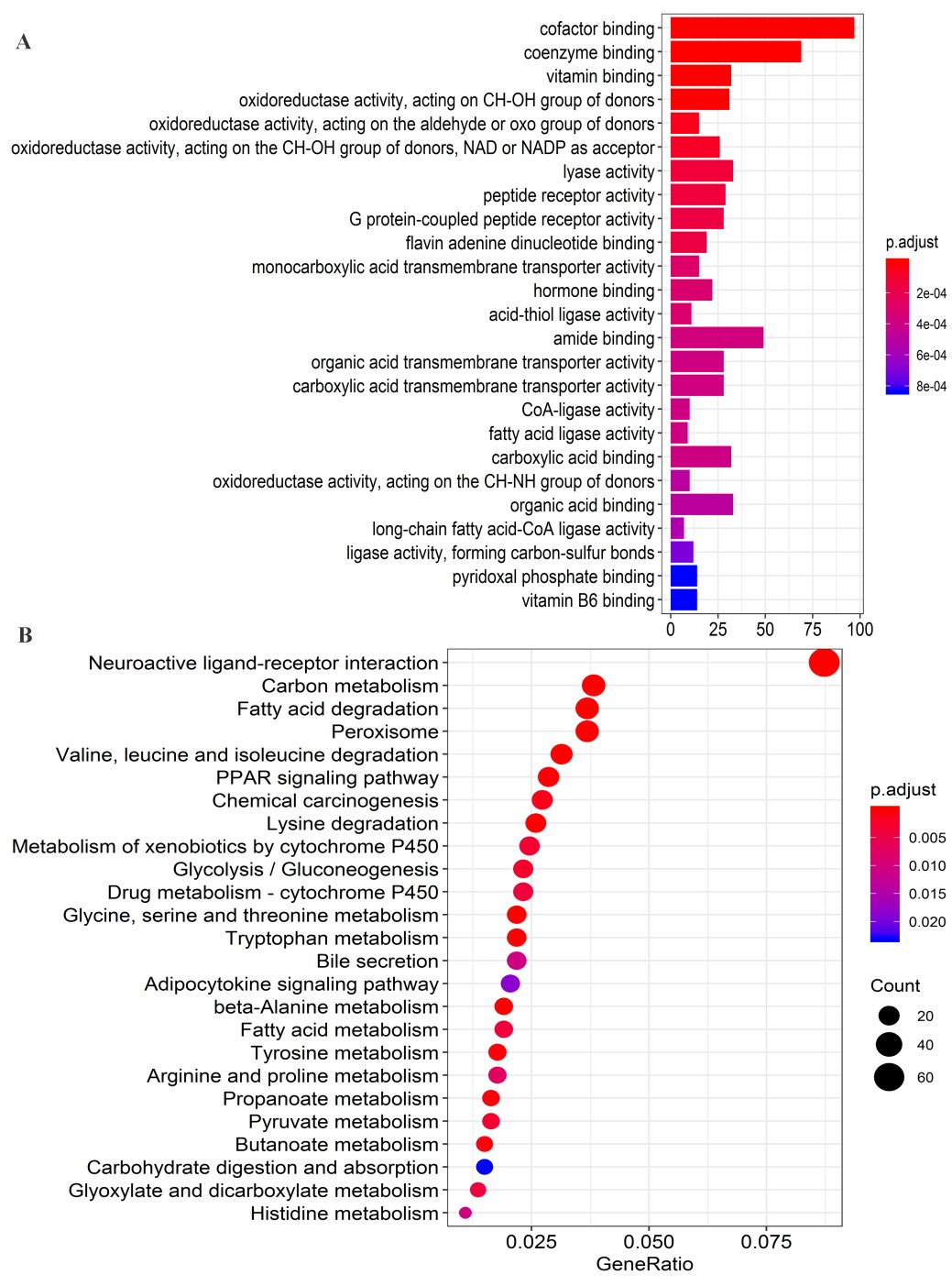

**Figure 8** **Functional enrichment analysis of four lncRNAs-associated protein-coding genes.** (A) Significantly enriched top 25 GO(gene ontology) terms. (B) Top 25 KEGG (Kyoto Encyclopedia of Genes and Genomes) pathways.

(PPAR) signaling pathway, amino acid metabolism, fatty acid metabolism, and chemical carcinogenesis.

## DISCUSSION

Currently, the popularity and wide application of gene microarray technology has presented considerable convenience to researchers who attempt to develop prognostic biomarkers of tumors. A large number of studies have confirmed that lncRNAs play an indispensable role in the proliferation, metastasis, metabolic regulation, and drug resistance of various tumors (*Pan et al., 2016*; *Kia et al., 2019*; *Liu et al., 2019*; *Schwarzenbacher et al., 2019*; *Zheng et al., 2019*). Therefore, their performance in determining the prognosis of various tumors has drawn widespread interest. The significance of prognostic signatures based on lncRNA expression has been demonstrated in various tumors, including those of head and neck squamous cell carcinoma (*Cao et al., 2017*), gastric cancer (*Song et al., 2017*), breast cancer (*Li et al., 2018*), cervical squamous cell carcinoma (*Mao et al., 2018*), and HCC (*Song et al., 2017*). For example, RGMB-AS1 plays an anti-tumor role by regulating a variety of biological processes in HCC cells (*Sheng et al., 2018*). Many unknown functional lncRNAs are yet to be explored. Moreover, cirrhosis of the liver is often associated with liver cancer, and more than 80% of patients with liver cancer have cirrhosis (*Affo, Yu & Schwabe, 2017*). To explore new prognostic lncRNAs in patients with cirrhotic HCC, patients with HCC and liver cirrhosis were included as subjects of the present study, and the lncRNA expression profile and clinical information were extracted from the GEO database for a comprehensive analysis.

We developed a prognosis formula for cirrhotic HCC based on the four lncRNAs, including AC093797.1, AL121748.1, AL162231.4, and POLR2J4, and verified it in the testing set. Patients were divided into high-risk and low-risk groups according to the prognostic signature score. Kaplan–Meier analysis confirmed that the four-lncRNA signature has favorable OS and RFS prediction ability and could be considered a new prognostic biomarker. Stratified analysis was used to assess the prognostic performance of this signature in patients with different clinical characteristics. We found that the four-lncRNA signature was significantly correlated with OS and RFS in patients with specific characteristics, including TNM stages I–II, solitary tumors, and BCLC stages 0–A. The selection of appropriate treatment methods is of prime importance for improving the prognosis of patients with early liver cancer (*Vitale et al., 2017*). In addition, the results of a relevant meta-analysis showed that liver transplantation had a higher OS rate and lower recurrence rate than hepatectomy among patients with liver cancer and Child-Turcotte-Pugh class A cirrhosis (*Zheng et al., 2014*). Although liver transplantation can achieve a better prognosis, it is especially important to be selective in determining the candidates for this procedure, owing to a shortage of donor livers. Our four-lncRNA label can help clinicians to predict and stratify the prognosis of patients after surgery, and implement reasonable treatment programs.

Among the four lncRNAs, AC093797.1 and POLR2J4 were risk factors for liver cancer, whereas the other two were protective factors (AL121748.1 and AL162231.4). With the

exception of POLR2J4, the lncRNAs were deemed prognostic markers of liver cancer for the first time, to our knowledge. Their function in HCC is unclear, and we determined their potential biological function through function enrichment analysis of related proteins. The GO enrichment analysis showed that the four lncRNA-related protein-coding genes were mainly involved in lipid metabolism and glycolysis, including fatty acid ligase activity, long-chain fatty acid-CoA ligase activity, oxidoreductase activity, and CoA ligase activity, among other processes. The KEGG pathway analysis showed that most genes were enriched in fatty acid and amino acid metabolism. This indicates that the four lncRNAs may be potentially involved in tumorigenesis and the development of tumors through the regulation of metabolism. At present, tumor metabolism is one of the more popular topics in tumor research. Some studies have reported that lncRNAs can affect tumor metabolism through a variety of methods, such as by regulating oncogenes, tumor suppressor factors, and crucial transcription factors to affect tumor growth (*Liu et al., 2019*). Thus, lncRNAs are expected to be targets for future tumor treatment. The specific mechanism of the four lncRNAs in the present study in the regulation of tumor metabolism still needs to be further clarified in cell and animal experiments.

This study is a preliminary study that explored the prognosis of cirrhotic HCC. The data of only a single liver cancer cohort were included in the GEO database that was used to develop and verify the lncRNA signature. In addition, an independent external validation set of larger samples to verify the reliability of the model is lacking. Moreover, further experimental studies to verify the potential functions of the four lncRNAs are also lacking. Nevertheless, the four-lncRNA signature developed, based on the expression of lncRNA from the GEO database, has the potential to become a new and highly effective biomarker for patients with cirrhotic HCC. The present study lays the foundation for further exploration of prognostic markers in patients with early HCC.

## CONCLUSIONS

In summary, we identified four potential lncRNAs biomarkers associated with prognosis in cirrhotic HCC, and constructed a risk score model. Patients in the high-risk group had lower overall survival and relapse-free survival than those in the low-risk group. Based on stratification analysis, the prognostic risk model might have ability to predict OS and RFS in cirrhotic HCC patients with BCLC stages 0–A, solitary tumors, and Tumor, Node, Metastasis system (TNM) stages I–II. Moreover, the four new lncRNAs identified could be used as potential therapeutic targets.

### Funding
The authors received no funding for this work.

### Competing Interests
The authors declare there are no competing interests.

## Author Contributions

- Linkun Ma conceived and designed the experiments, performed the experiments, analyzed the data, contributed reagents/materials/analysis tools, prepared figures and/or tables, authored or reviewed drafts of the paper.
- Cunliang Deng conceived and designed the experiments, authored or reviewed drafts of the paper, approved the final draft.

## Data Availability

Data is available at NCBI GEO under accession number GSE14520.

## Supplemental Information

Supplemental information for this article can be found online at http://dx.doi.org/10.7717/peerj.7413#supplemental-information.

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
