# Peer review of "Identification of a novel four-lncRNA signature as a prognostic indicator in cirrhotic hepatocellular carcinoma"

_PeerJ, doi:10.7717/peerj.7413_

## Round 0.1 · original submission · Major Revisions

Please address the points raised by the reviewers.
In addition to that please rewrite the legend for Figure 1. It is not in the standard format.
The authors in their conclusion made the statement "The results of functional prediction showed that four lncRNAs may be closely related to amino acid, lipid, and glucose metabolism in cirrhotic HCC". These are strong conclusions just based on predictions. So they should tone-down this by warranting functional studies and replication.

Reviewer 1 ·

Basic reporting

The manuscript was mostly clear with professional use of language. I have a few grammatical suggestions:
1. Line 41: “Liver Cirrhosis is an important prognostic risk factor in patients with liver cancer”.
2. Line 49: please omit “a total of 204”. It is understood that the 204 cirrhotic HCC samples came from 204 patients, unless otherwise stated.
3. Line 75: please remove “as a highly fatal disease”. The next sentence reiterates that point.

Sufficient background provided barring a few references:

1. Line 75: please provide statistics for liver cancer related deaths, with references
2. Line 76: please provide statistics for increasing mortality and why “new interventions” have become absolutely necessary.
3. Line 101: Since the study is based on lncRNAs as prognostic indicators in HCC, it would be great if the authors could cite an example of a lncRNA used for patient stratification.

Professional article structure, figures and tables.

Experimental design

The article falls within the scope of the journal. The introduction is well laid out with a description of liver cancers and HCCs. It also sets about addressing the association between liver cirrhosis and the risk of developing HCC. It then goes on to explain why lncRNAs may be important clinical indicators and therapeutic targets for patients with liver cancer. However, the purpose of this study is unclear, thus It would be useful to include a sentence or two about “aims” or “objectives” of the study at the end of the introduction.

It is also unclear how this particular study is addressing a knowledge gap. The authors need to explicitly state what the knowledge gap is i.e. the stratification of HCC is poorly done and therefore the authors wish to add a more robust prognostic strategy or no one has studied cirrhosis as a prognostic marker in HCC and therefore the authors endeavor to establish that association?

Materials and methods are generally well written with enough information to allow replication of study. Here are a few comments:
1. How was the correlation coefficient cutoff of 0.4 determined? R = 0.4 seems like a low threshold for association.
2. Some of the material included in results should be moved to Methods. For example, lines 190 – 193 describe patient stratification using a software. This can be moved to methods as the authors are not reporting any findings here. Lines 201 – 207 talk about cinstructing prognostic signatures and calculating risk scores. This can be moved to Methods as the authors are still describing steps that they used to get to their results.
3. Lines 205-207: what are the numbers/coefficients -0.4818, 0.4404, 1.2845 and -1.5170 in the risk score calculation? How were these numbers derived?

Validity of the findings

Data seems statistically sound and analysis appears to be robust, with the conclusion well stated. The original questions needs to be well defined, together with the aim of the study and then the conclusion needs to be restated in that context, if necessary.

·

Basic reporting

Summary: The authors identify a novel four-lncRNA signature as a prospective candidate biomarker in cirrhotic hepatocellular carcinoma through a publicly available dataset that they analysed. A combination of machine learning heuristics and statistics are supplemented with their results.

Strengths: The authors identify perhaps a very important dataset and come up with a pipeline that could simply be based on an in silico approach

Weaknesses: The narration could have been nicer. A few important references could be a good addition besides having a methodological flowchart.

Experimental design

The rationale of the work appeared to be lost. On a hindsight, the authors MUST describe why lncRNA signatures, after all.

Moving further to methods, the dataset the authors considered could have some textual insights like population, ethnicity etc., They could describe in a sentence or two pointing to the erstwhile tables they cited.

To the end, they could also subtly point what if there are RNA-Seq datasets that they could consider.

Although Kaplan-Meier survival curve is drawn through R, a better reference must be pointed out in lines 160-165.

A good flowchart for methods would be very nice addition.

The Pearson coefficient between 0.2 and 0.4 with moderate effect could have been considered. How good is this justifiable ( >0.4) ?

Lines 189-194 could move to Materials and methods

Validity of the findings

The lncRNAs were not checked for curation/bona fidelity. Did the authors verify them?

For example, they could have subtly checked for their role in NONCODE etc.

Although in lines 226-230, the authors mention the cut-off values for training and test datasets, arbitrary division of test datasets is perhaps not a good idea. It might lead to bias, if they are divided based on high/low risk on the test datasets as well. How good was the precision/recall in this case? Did the authors check this?

When the SRA dataset is mapped to the genes, PLPBP is shown to be a candidate protein-coding gene associated with tumorigenesis of HCC. Did the authors check for lncRNA-mRNA interactions. A gamut of tools for such predictions do exist. This would have been an awesome to come up with their role for biomarker analyses.

Where are these signatures localised? To which chromosome? A pictorial representation would be very good.

Additional comments

Line 47: The word 'dataset' could be one word
Please check typos all through and a few words are capitalised in between

Line 75: Introduction could have Globocan statistics as a reference with thorough HCC prevalence details.

Lines 90-104: A good number of works in the recent past have dealt identification of lncRNAs as candidate biomarkers. For example., Hu et al., 2017; Yuan et al., 2017. These authors have also discussed on the role of circulating biomarkers towards prognosis/diagnosis making a good way to consider the role of ncRNAs,viz. SVUGP2/GIHCG etc.

These sections could have a very good rationale to start with. For example., why lncRNA signatures alone? The role of miRNAs and their respective interaction studies with mRNA etc. An absolute rationale for this study is missing.

The article must be proofread thoroughly in the next version.

The authors contributions and funding source etc., may be added.

---

## Round 0.2 · accepted · Accept

With the implementation of reviewers and internal comments, the manuscript´s quality is increased and acceptable for publication in PeerJ.

Reviewer 1 ·

Basic reporting

The authors have addressed my concerns satisfactorily and I do not have any further comments and inputs to add

Experimental design

The authors have addressed my concerns satisfactorily and I do not have any further comments and inputs to add

Validity of the findings

The authors have addressed my concerns satisfactorily and I do not have any further comments and inputs to add

Additional comments

The authors have addressed my concerns satisfactorily and I do not have any further comments and inputs to add

·

Basic reporting

I am satisfied with all the comments rendered.Thank you.

Experimental design

I am satisfied with all the comments rendered

Validity of the findings

I am satisfied with all the comments rendered.Thank you.

Additional comments

I am satisfied with all the comments rendered.Thank you.